# The target of the DEAH-box NTP triphosphatase Prp43 in *Saccharomyces cerevisiae* spliceosomes is the U2 snRNP-intron interaction

Jean-Baptiste Fourmann[1], Olexandr Dybkov[1], Dmitry E Agafonov[1], Marcel J Tauchert[2], Henning Urlaub[3,4], Ralf Ficner[2], Patrizia Fabrizio[1]*, Reinhard Lührmann[1]*

[1]Department of Cellular Biochemistry, Max-Planck-Institute for Biophysical Chemistry, Göttingen, Germany; [2]Department of Molecular Structure Biology, Institute for Microbiology and Genetics, Georg August University of Göttingen, Göttingen, Germany; [3]Bionalytics, Institute for Clinical Chemistry, University Medical Center Göttingen, Göttingen, Germany; [4]Bioanalytical Mass Spectrometry Group, Max Planck Institute for Biophysical Chemistry, Göttingen, Germany

**Abstract** The DEAH-box NTPase Prp43 and its cofactors Ntr1 and Ntr2 form the NTR complex and are required for disassembling intron-lariat spliceosomes (ILS) and defective earlier spliceosomes. However, the Prp43 binding site in the spliceosome and its target(s) are unknown. We show that Prp43 fused to Ntr1's G-patch motif (Prp43_Ntr1GP) is as efficient as the NTR in ILS disassembly, yielding identical dissociation products and recognizing its natural ILS target even in the absence of Ntr1's C-terminal-domain (CTD) and Ntr2. Unlike the NTR, Prp43_Ntr1GP disassembles earlier spliceosomal complexes (A, B, B[act]), indicating that Ntr2/Ntr1-CTD prevents NTR from disrupting properly assembled spliceosomes other than the ILS. The U2 snRNP-intron interaction is disrupted in all complexes by Prp43_Ntr1GP, and in the spliceosome contacts U2 proteins and the pre-mRNA, indicating that the U2 snRNP-intron interaction is Prp43's major target.

*For correspondence: Patrizia. Fabrizio@mpi-bpc.mpg.de (PF); Reinhard.Luehrmann@mpi-bpc. mpg.de (RL)

**Competing interests:** The authors declare that no competing interests exist.

## Introduction

The spliceosome assembles de novo for each new round of pre-mRNA splicing by the sequential recruitment of the U1, U2 and U4/U6.U5 small nuclear ribonucleoproteins (snRNPs) and numerous non-snRNP proteins to the intron. It undergoes a series of remodeling steps that are driven by eight conserved DExD/H-box ATPases or RNA helicases (*Cordin et al., 2012*; *Cordin and Beggs, 2013*). Initially, U1 snRNP recognizes and forms a base-pairing interaction with the 5' splice site (SS) of the intron. Recruitment of the U2 snRNP to the branch-site (BS) yields the A complex. U2 snRNA forms a base-pairing interaction with the BS nucleotides, while several U2 SF3a and SF3b proteins stabilize U2 snRNP binding to the intron by interacting with intron nucleotides upstream and downstream of the BS (*Gozani et al., 1996*; *Schneider et al., 2015*). The U4/U6.U5 tri-snRNP associates with the A complex, and a short helix between the 5' end of U2 and 3' end of U6 snRNA (U2/U6 helix II) is formed. Subsequently, the U1/5'SS interaction is disrupted, allowing base-pairing of the 5'SS with U6 snRNA, resulting in the formation of the pre-catalytic B complex. Activation of the spliceosome, yielding the B[act] complex, involves unwinding of the U4/U6 duplex by the RNA helicase Brr2, and displacement of U4 snRNA from the spliceosome. This allows U6 snRNA to interact with U2 snRNA

to form U2/U6 helix I. The resulting U2/U6/pre-mRNA interaction network forms the heart of the spliceosome's catalytic center (*Datta and Weiner, 1991*; *Wu and Manley, 1991*; *Madhani and Guthrie, 1992*). Concomitant with these RNA rearrangements, more than twenty new proteins, including the Prp19 complex (NTC), are stably integrated into the $B^{act}$ complex (*Fabrizio et al., 2009*).

For catalytic activation, the $B^{act}$ complex is remodeled by the RNA helicase Prp2 in cooperation with its G-patch protein co-factor Spp2, yielding the B* complex (*Kim and Lin, 1996*; *Roy et al., 1995*; *Silverman et al., 2004*; *Warkocki et al., 2009*; *2015*), in which step 1 catalysis occurs, whereby the 2' hydroxyl group of the BS adenosine attacks the 5'SS, generating the cleaved-off 5' exon and the intron-lariat (IL)-3'exon intermediates (*Chiu et al., 2009*; *Warkocki et al., 2009*). The newly formed C complex catalyzes step 2 of splicing, leading to exon ligation (*Horowitz, 2012*). The mature mRNA is then dissociated from the intron-lariat spliceosome (ILS).

The ILS is dismantled by the DEAH-box NTPase Prp43, which is necessary for recycling snRNPs and splicing factors, and thereby promotes efficient splicing in the cell (*Arenas and Abelson, 1997*; *Martin et al., 2002*; *Tsai et al., 2005*; *Fourmann et al., 2013*). In yeast, Prp43 associates with two co-factors to dismantle the ILS, the G-patch protein Ntr1 (also termed Spp382) and Ntr2, which together form the NTR complex (*Boon et al., 2006*; *Pandit et al., 2006*; *Tsai et al., 2007*). Ntr1 binds through its N-terminal G-patch motif (GP) to the C-terminal domain of Prp43 and thereby stimulates Prp43's ATPase and RNA helicase activity (*Tanaka et al., 2007*; *Christian et al., 2014*). Their productive interaction in the NTR is also required to dissociate the ILS in yeast splicing extracts, where Ntr2 is thought to recruit Ntr1 and thus also Prp43 (*Tsai et al., 2005*; *2007*; *Tanaka et al., 2007*).

In a purified yeast splicing system, Prp43 and its co-factors are sufficient to dissociate not only the intron lariat (IL) RNA from the ILS, but also the snRNPs from each other, yielding free U6 snRNA, a 20S U2 and an 18S U5 snRNP (*Fourmann et al., 2013*). Thus, while the effects of NTR action on the ILS and the nature of the dissociation products have been well characterized, the mechanism how Prp43 dismantles the ILS and, most importantly, the target structure(s) that are disrupted (remodeled) by Prp43 in the spliceosome are completely unknown. As the complex RNA interaction network formed by the intron and U2/U6 snRNAs is still in place in the post-catalytic ILS (*Chan et al., 2003*; *Yan et al., 2015*), several RNA duplexes, including the U2/BS and U2/U6 helices I and II must be dissociated during ILS disassembly by Prp43. This raises the question whether Prp43 has several RNA duplex targets in the ILS and if so, whether it dissociates them in a sequential manner. Moreover, the ILS is a large RNP complex and DEAH-box ATPases can also act as RNPases (*Jankowsky et al., 2001*). Thus the target(s) of Prp43 in the ILS may also involve protein–RNA interactions.

Identifying Prp43's target structure(s) is also of special interest as Prp43 is not only required for ILS disassembly, but is also implicated in the discard pathway of spliceosomes associated with sub-optimal pre-mRNA substrates, which become stalled at earlier stages of the spliceosomal cycle (*Pandit et al., 2006*; *Semlow and Staley, 2012*; *Koodathingal and Staley, 2013*). Thus, Prp43 is also involved in the disassembly of early spliceosomes that not only differ significantly from the ILS in their biochemical composition, but also in the nature of their RNA interaction network. The role of Ntr2 and the C-terminal domain (CTD) of Ntr1 in facilitating Prp43's recognition of its spliceosomal target structures, is also not well understood. It has been suggested that Ntr2 may help to recruit Ntr1 and Prp43 to the ILS (*Tsai et al., 2007*). Alternatively, or in addition, they may also play a role in regulating the productive interaction of Ntr1's GP with Prp43 in the spliceosome and help to prevent dissociation of properly assembled spliceosomes other than post-catalytic ILSs.

To address these questions, we first set out to define the minimal factor requirements for Prp43 to efficiently dismantle purified ILSs in vitro. We show that a fusion protein of Prp43 and the G-patch motif of Ntr1 (Prp43_Ntr1GP) efficiently dismantles the ILS, yielding the same dissociation products as the NTR. Thus, Prp43_Ntr1GP is a disassembly factor that recognizes its natural target structure(s) in the ILS even in the absence of Ntr1's CTD and Ntr2. Prp43_Ntr1GP also efficiently dismantles activated ($B^{act}$) spliceosomes, while NTR does not. This indicates that Ntr2 and Ntr1's CTD act as a 'doorkeeper' that prevents dissociation by NTR of properly assembled spliceosomes other than ILSs. Prp43_Ntr1GP also dissociates purified B and A complexes. Characterization of the Prp43_Ntr1GP-mediated dissociation products of the $B^{act}$, B and A complexes revealed that in all complexes U2 snRNP is disloged from the pre-mRNA, indicating that the interaction of U2 snRNP with the

extended branch site is the major target of Prp43 in the spliceosome. The latter is further supported by our finding that Prp43_Ntr1GP binds to the pre-mRNA and contacts predominantly U2 proteins in the spliceosome. Our data are consistent with the idea that Prp43 specifically displaces U2 snRNP from the BS by moving along the intron in a 5′ to 3′ direction or acting locally by disrupting RNA structures without significant translocation.

## Results

### Ntr1's G-patch motif is sufficient to promote dissociation of purified ILSs in the same manner as the NTR

Previously, it was shown that a fusion protein, in which the C-terminal amino acid of Prp43 is connected to the N-terminal amino acid of the Ntr1 G-patch (termed Prp43_Ntr1GP, *Figure 1A*), also displays high RNA-stimulated ATPase and RNA helicase activities (*Christian et al., 2014*). Therefore we investigated whether Prp43_Ntr1GP is also sufficient to dismantle the ILS and thus represents a minimal spliceosome disassembly system. Purified ILSs were obtained as described previously (*Figure 1—figure supplement 1*) (*Fourmann et al., 2013*). As shown in *Figure 1B*, incubation of the ILS with ATP and recombinant Prp43 plus Ntr1 and Ntr2 (forming the NTR complex, see *Figure 1A*) (*Tsai et al., 2007*) resulted in the release of ∼80% of the IL RNA and dissociation of the remaining spliceosomal RNP core into free U6 snRNA, 18S U5 and 20S U2 snRNPs, while in the absence of the NTR (*Figure 1C*) or in the presence of Prp43 alone (*Figure 1D*), only a very low level of the ILS was released (*Fourmann et al., 2013*). Remarkably, disassembly of the ILS by Prp43_Ntr1GP yielded the same snRNP and RNA dissociation products (i.e., IL RNA, free U6 snRNA, 18S U5 and 20S U2 snRNPs) (*Figure 1E*) as those obtained when the NTR was added (*Figure 1B*). Addition of Ntr2 to the reaction mixture did not influence the efficiency of ILS disassembly by Prp43_Ntr1GP (*Figure 1F*), indicating that Ntr2 and the C-terminal domain of Ntr1 are not needed for recruiting Prp43_Ntr1GP to the spliceosome. In summary, these results demonstrate that Ntr1's GP is essential and sufficient to promote ILS disassembly by Prp43 in the same manner as the NTR complex. This in turn suggests that Prp43_Ntr1GP recognizes the same target structure(s) in the ILS as the NTR.

### Prp43_Ntr1GP, but not the NTR, disassembles B$^{act}$ complexes

Our findings indicate that Prp43_Ntr1GP represents a minimal system for ILS disassembly. We next investigated whether Prp43_Ntr1GP can also dismantle other spliceosomal complexes, including those that are not disassembled by the NTR. Consistent with previous results (*Chen et al., 2013*), purified B$^{act\ \Delta Prp2}$ complexes (lacking Prp2) that were assembled in yeast extracts on a wild-type actin7 (Act7-wt) pre-mRNA, remained stable during incubation with the NTR and ATP, compared with ATP alone (*Figure 2A,B*) or Prp43 alone and ATP (data not shown). In striking contrast, Prp43_Ntr1GP dissociated B$^{act\ \Delta Prp2}$ complexes with high efficiency, yielding free U6 snRNA that migrates on top of the gradient, a 20S U2 snRNP, and a U5 snRNP that displays a broad migration behavior with a peak at ∼35S (*Figure 2C*), indicating that all snRNPs were separated from each other (*Figure 2C*).

The bulk of pre-mRNA appeared to co-migrate in the gradient with the 35S U5 snRNP but not with the U2 snRNP (*Figure 2C*), indicating that the latter was no longer bound to the pre-mRNA. Indeed, immunoprecipitation (IP) experiments carried out with antibodies specific for the U5 snRNP protein Snu114 and fractions across the gradient revealed that pre-mRNA co-precipitates together with the U5 snRNP; while very little or no U2 and U6 snRNAs were found in the IPs (*Figure 2D*, lanes 4 and 5). Interestingly, the U5 snRNA and pre-mRNA (but not U2 and U6 snRNAs), were also co-precipitated from the gradient with anti-Prp19 antibodies (*Figure 2D*, lanes 8 and 9), suggesting that the NTC proteins are bound to the U5 snRNP after disassembly of the B$^{act\ \Delta Prp2}$ complex. This conclusion is also supported by mass spectrometry (MS) analyses of the purified 35S U5-pre-mRNA complex, showing that the majority of NTC proteins migrated in fractions 13–16, together with U5 proteins (*Figure 2—source data 1*). Finally, a more detailed characterization of the disassembly products of the B$^{act\ \Delta Prp2}$ complex by UV protein-RNA crosslinking and MS revealed that all U2 SF3a and SF3b proteins, which contact the BS region of the intron within the B$^{act}$ complex (*Schneider et al., 2015*), are dissociated from the intron and remain associated with the released 20S U2 snRNP (*Figure 2E—source data 1*). Indeed, we found that Hsh155 (human SF3B1/SAP155)

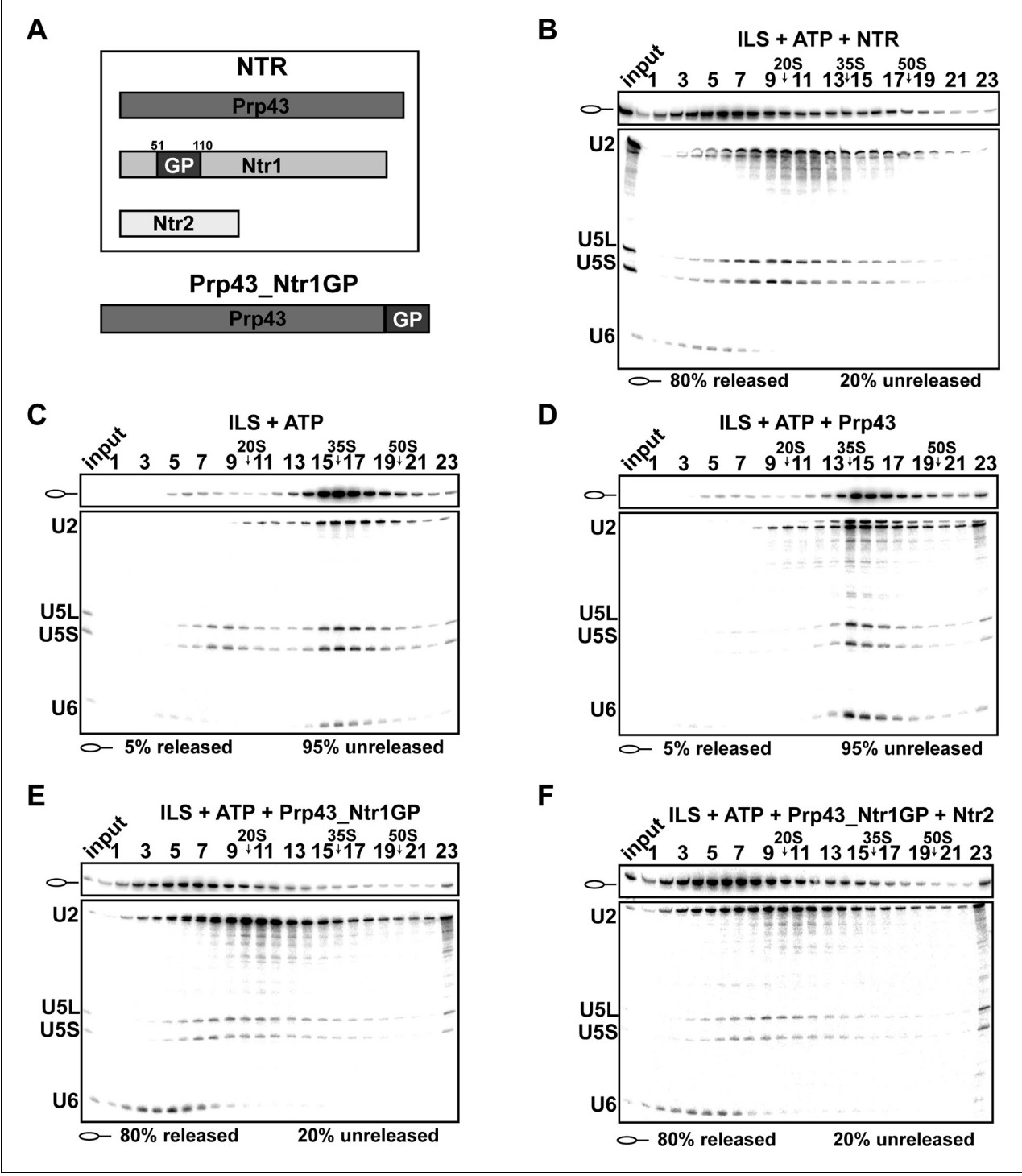

**Figure 1.** Prp43 fused to the G-patch motif of Ntr1 is sufficient to promote dissociation of the ILS. (**A**) Upper panel, schematic representation of the NTR complex, composed of Prp43, Ntr1 and Ntr2; lower panel, Prp43_Ntr1GP in which the G-patch of Ntr1 is fused to the C-terminal domain of Prp43. (**B**) 10–30% glycerol gradient sedimentation of purified ILS incubated in solution with ATP plus NTR, (**C**) no recombinant protein, (**D**) Prp43 (**E**) Prp43 fused to Ntr1GP (Prp43_Ntr1GP), or (**F**) Prp43_Ntr1GP and Ntr2. U2, U5 and U6 snRNAs were visualized by Northern blotting followed by autoradiography. RNA identities are indicated on the left. Quantifications were performed with ImageQuant software (Molecular Dynamics, Pittsburg, PA). Numbers represent the percentage of intron-lariat RNA released in the top fractions (sum of fractions 1–11) or associated with the ILS (unreleased, sum of fractions 12–23) relative to the intron-lariat RNA distributed in all 23 fractions, the sum of which was set to 100%.

The following figure supplement is available for figure 1:

Figure 1 continued

**Figure supplement 1.** Isolation of intron-lariat spliceosomes (ILSs).

can be crosslinked to the pre-mRNA in the intact $B^{act\ \Delta Prp2}$ complex (*Figure 2E*, lane 2), but after $B^{act\ \Delta Prp2}$ disassembly, crosslinking of Hsh155 to the pre-mRNA was no longer observed (lane 4). This indicates that Hsh155 is completely displaced from the pre-mRNA by Prp43_Ntr1GP. To validate the quantitative dissociation of the U2 snRNP from the pre-mRNA, we carried out IP experiments using the Cus1-TAP extract (human SF3B145/SAP145). Disassembled $B^{act\ \Delta Prp2}$ spliceosomes were incubated with Protein A–Sepharose beads that carried IgG, to which the TAP tag binds efficiently. *Figure 2F* (lanes 3,4) shows a selective immunoprecipitation of the U2 snRNP, indicating quantitative dissociation of the U2 particle from the pre-mRNA, which is thus not coprecipitated together with the U2 snRNP. In conclusion, Prp43_Ntr1GP dismantles the $B^{act\ \Delta Prp2}$ complex and ILS in a similar manner, suggesting that it also recognizes similar target structure(s) in both complexes (see also Discussion below). Furthermore, the striking finding that Prp43_Ntr1GP, but not the NTR, efficiently dismantles the $B^{act\ \Delta Prp2}$ complex, suggests that in the $B^{act\ \Delta Prp2}$ complex the CTD of Ntr1 and/or Ntr2 negatively regulate Prp43 and prevent disassembly of an activated spliceosome which is assembled on a wild-type pre-mRNA.

## The U2 snRNP-intron interaction is a major target of Prp43 in the spliceosome

As Prp43_Ntr1GP, in contrast to the NTR, appears to disassemble spliceosomes in an unconstrained manner, we next investigated whether it is also capable of dismantling pre-catalytic B complexes. The RNA interaction network within the B complex differs in several respects from that in the $B^{act}$ and ILS complexes (*Figure 3A*), and thus the nature of B complex dissociation products may provide valuable insight into the target structure(s) that is/are recognized by Prp43 in the spliceosome.

B complexes were generated by incubating Act7-wt pre-mRNA with splicing extracts in the presence of 50 µM ATP to prevent activation of the spliceosome by the Brr2 RNA helicase. Brr2 is responsible for U4/U6 snRNAs unwinding, a critical step in spliceosomal activation, which can only occur at higher ATP concentrations (i.e. ~100–200 µM ATP) (*Cheng, 1994*; *Fabrizio et al., 2009*). To exclude any effect of Brr2 RNA helicase, which is strictly ATP-dependent, we incubated purified B complexes either with the NTR or with Prp43_Ntr1GP in the presence of UTP. The vast majority of B complexes remained stable during incubation with UTP alone, and addition of the NTR did not lead to a significant increase in dissociation (*Figure 3B and C*). In contrast, in the presence of Prp43_Ntr1GP and UTP (*Figure 3D*), 67% of the B complexes were disassembled. One third of U2 snRNA co-migrated during glycerol-gradient centrifugation with U4, U5 and U6 snRNAs (peaking in fractions 13–15), but clearly separated from the pre-mRNA, which peaked in fractions 9–11 (*Figure 3D*). The remaining fraction of U2 snRNA migrated in fractions 7–11, while U1 snRNA exhibited a broad peak in fractions 10–15. Importantly, and in contrast to the disassembly of $B^{act}$ complexes, only negligible amounts of U6 snRNA were found in the top fractions of the gradient (*Figure 3D*).

To investigate which snRNPs are still associated with each other and/or with pre-mRNA after the action of Prp43_Ntr1GP, we first performed IPs with anti-Snu114 antibodies and fractions across the gradient. Immunoprecipitates from fractions 13–15 contained similar amounts of U2, U4, U5 and U6 snRNAs, but only minor amounts of pre-mRNA and U1 (*Figure 3E*, lane 5), indicating that predominantly a U2.U4/U6.U5 tetra-snRNP was present in these fractions, consistent with its Svedberg value of ~35S (*Gottschalk et al., 1999*). Immunoprecipitates from fractions 10–12 contained similar amounts of U4, U5 and U6 snRNAs but little U1 and U2 snRNAs, and low amounts of pre-mRNA, indicating the presence of predominantly 25S U4/U6.U5 tri-snRNPs (*Figure 3E*, lane 4). The presence of low amounts of U1 in both immunoprecipitates suggests that the U1 snRNP, which is present in substoichiometric amounts in purified B complexes, but no longer base paired to the 5'SS (*Fabrizio et al., 2009*), may be loosely-associated with the tri- and tetra-snRNPs in these fractions.

MS analysis of the glycerol gradient fractions was consistent with the presence of intact U2, 25S U4/U6.U5 tri-snRNPs and 35S U2.U4/U6.U5 tetra-snRNPs in the corresponding fractions (*Figure 3— source data 1*). MS also revealed a bipartite distribution of the NTC proteins which co-migrated

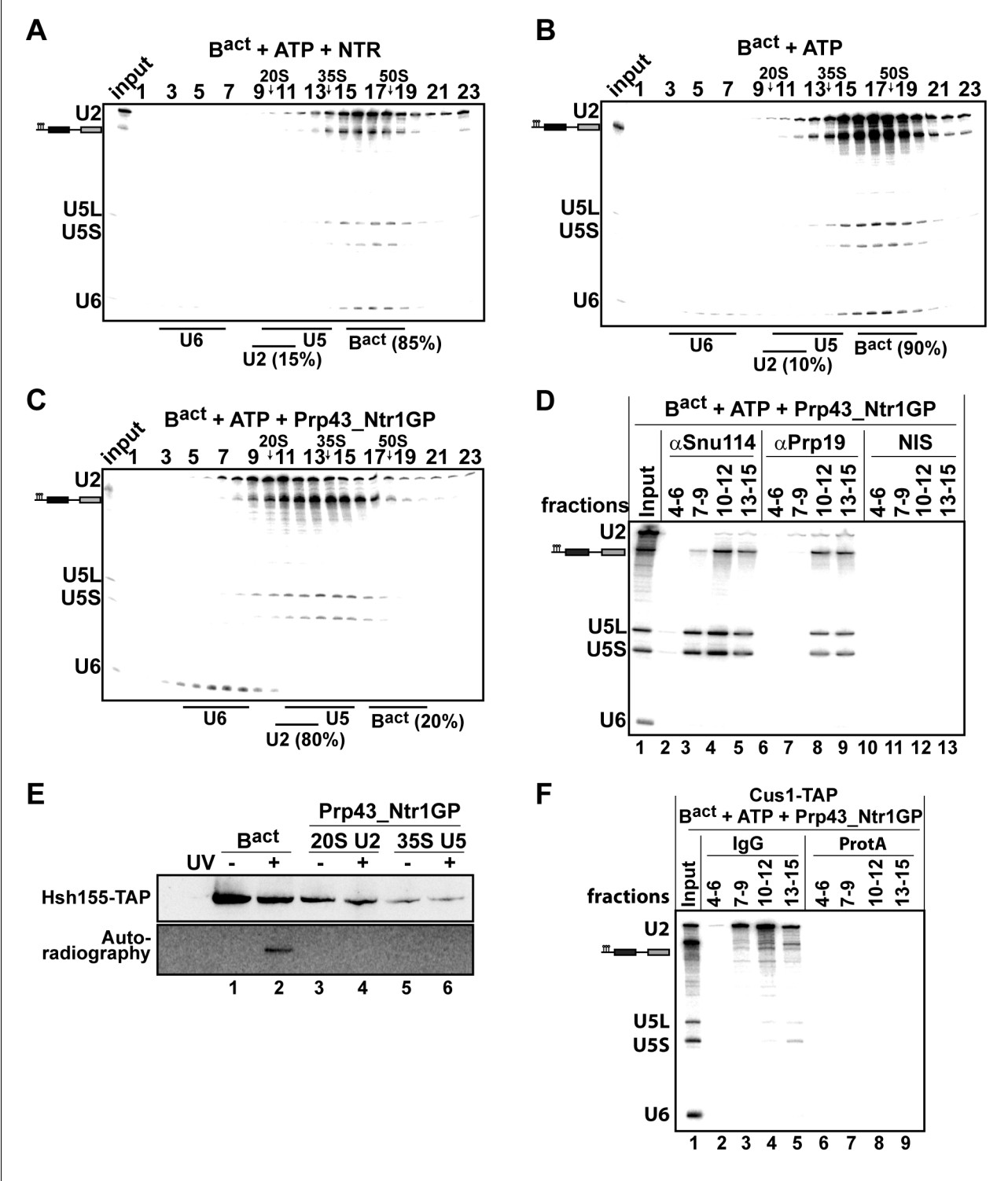

**Figure 2.** Prp43_Ntr1GP but not NTR disassembles B$^{act}$ complexes formed on wild-type actin7 pre-mRNA. 10–30% glycerol gradient sedimentation of purified B$^{act\ \Delta Prp2}$ complexes incubated with ATP and (**A**) NTR, (**B**) no recombinant protein, or (**C**) Prp43_Ntr1GP. Samples were analyzed as described in *Figure 1*. Quantifications were performed with ImageQuant software (Molecular Dynamics). Numbers represent the percentage of U2 snRNA released in the top fractions (sum of fractions 1–11) or associated with the B$^{act\ \Delta Prp2}$ complex (unreleased, sum of fractions 12–23) relative to the U2 snRNA distributed in all 23 fractions, the sum of which was set to 100%. (**D**) B$^{act\ \Delta Prp2}$ spliceosomes were incubated with ATP and Prp43_Ntr1GP. Five% of the input was withdrawn (lane 1). Samples were separated on 10–30% glycerol gradients. Every three fractions from 4 to 15 (4–6, 7–9, etc.) were combined and immunoprecipitated with anti-Snu114, anti-Prp19 antibodies or with non-immune serum (NIS). Co-precipitated RNAs were analysed by Northern

*Figure 2 continued on next page*

*Figure 2 continued*
blotting using probes that hybridize to U2, U5 and U6 RNA. (E) B$^{act\ \Delta Prp2}$ spliceosomes harboring Hsh155-TAP were assembled on uniformly radiolabeled pre-mRNA. One half was left untreated and the other half was incubated with Prp43_Ntr1GP and ATP, and then analysed on separate glycerol gradients. Peak fractions corresponding to intact B$^{act\ \Delta Prp2}$ (15–18), 20S U2 (9–11) and 35S U5 (13–15) were UV-irradiated or untreated. Proteins and protein-RNA crosslinks were separated by SDS-PAGE and transferred to a membrane. Western blot analysis was performed using PAP complex antibodies. Protein crosslinked to radiolabeled pre-mRNA nucleotides was visualized by autoradiography of the same membrane. (F) B$^{act\ \Delta Prp2}$ complexes containing the U2 protein Cus1 tagged with the TAP tag (Cus1-TAP) were incubated with ATP and Prp43_Ntr1GP and separated on a glycerol gradient. The indicated three gradient fractions were combined and immunoprecipitated with IgG Sepharose beads (IgG) allowing the selective immunoprecipitation of 20S U2 snRNP (lanes 3,4). The same fractions were also precipitated with Protein A–Sepharose beads (ProtA) (lanes 6–9). Co-precipitated snRNAs were analyzed on an 8% urea-polyacrylamide gel and identified by Northern blotting as above.
The following source data is available for figure 2:

**Source data 1.** Protein composition of B$^{act\ \Delta Prp2}$ complexes disassembled by Prp43_Ntr1GP and ATP.

with the pre-mRNA and the tetra-snRNP (*Figure 3—source data 1*). Upon IP with anti-Prp19 antibodies, significant amounts of pre-mRNA – but only minor amounts of U2 and the other snRNAs – were precipitated from fractions 10–12 (*Figure 3E*, lane 8), whereas tetra-snRNPs were precipitated from fractions 13–15 (lane 9). This indicates that i) NTC proteins bind directly to pre-mRNA – which may also explain why the pre-mRNA migrates with an S value of ~20S and not in the top fractions of the gradient (*Figure 3D*) – and ii) the U2 snRNPs migrating in fractions 9–11 (*Figure 3D*) are not bound to the pre-mRNA.

The latter was supported by UV RNA-protein crosslinking experiments. That is, the U2 protein Prp9 (human SF3A60/SAP61) could not be crosslinked to pre-mRNA in fractions 9–11 (i.e 20S U2) of the gradient containing B complexes disassembled by Prp43_Ntr1GP and UTP, while in intact B complexes it was crosslinked (*Figure 3F*, compare lanes 2 and 4). Likewise, no crosslinks between Prp9 and pre-mRNA were observed in the U2.U4/U6.U5 tetra-snRNP peak fractions (lane 6). Pull-downs using IgG-Sepharose and the Cus1-TAP extract confirmed that the pre-mRNA is not coprecipitated together with the U2 snRNP (*Figure 3G*, lanes 3,4), and only slightly from fractions 13–15 containing the U2.U4/U6.U5 tetra-snRNPs and some U2 and U4/U6.U5 tri-snRNPs (lane 5). We conclude that disassembly of B complexes by Prp43_Ntr1GP involves the complete displacement of U2 snRNP from the pre-mRNA BS region. U2 snRNP is very probably displaced as a tetra-snRNP particle, which partially dissociates into free 20S U2 snRNPs and 25S U4/U6.U5 tri-snRNPs during gradient centrifugation. Taken together, these results are consistent with the idea that Prp43_Ntr1GP disrupts primarily the U2 snRNP-BS interaction.

The earliest spliceosomal complex containing U2 snRNP stably bound to the pre-mRNA BS, is complex A. We therefore investigated whether Prp43_Ntr1GP can dislodge U2 snRNP from the pre-mRNA even at this early stage. The majority of the A complex remained intact after incubation with Prp43 and ATP (*Figure 4A*) or ATP alone (data not shown), so that the U1 and U2 snRNPs sedimented together with the pre-mRNA on the gradient as a ~37S complex. In contrast, after incubation with Prp43_Ntr1GP and ATP, the majority of U2 migrated in the 20S region of the gradient, whereas U1 and pre-mRNA co-migrated at ~27S (*Figure 4B*). IP of gradient fractions after incubation with Prp43_Ntr1GP and ATP, using an antibody against the U1 protein Nam8, showed significant co-precipitation of pre-mRNA with U1 snRNP, but only negligible co-precipitation of U2 snRNP in fractions 10–12 and 13–15 (*Figure 4C*, lanes 4 and 5). Thus, in contrast to the U1 snRNP, U2 snRNP is displaced from the pre-mRNA by Prp43_Ntr1GP. These results are supported by the distribution of proteins across the gradient after disassembly of the A complex with Prp43_Ntr1GP and ATP, as analyzed by MS (*Figure 4—source data 1*). Thus, these results support strongly that the main target of Prp43_Ntr1GP in the spliceosome is the U2 snRNP–BS interaction.

## Prp43_Ntr1GP interacts only with the pre-mRNA and not with U2 or any other snRNA in the B$^{act}$ complex

To narrow down Prp43's target site, we next investigated Prp43_Ntr1GP's RNA docking site in the spliceosome. The 3' end of the intron immediately downstream of the BS is essential for the Prp2 RNA helicase-mediated remodeling of the U2 SF3a/b proteins during the transformation of B$^{act\ \Delta Prp2}$ into catalytically activated B* complexes, and Prp2 binds to this RNA region (*Liu and Cheng, 2012*;

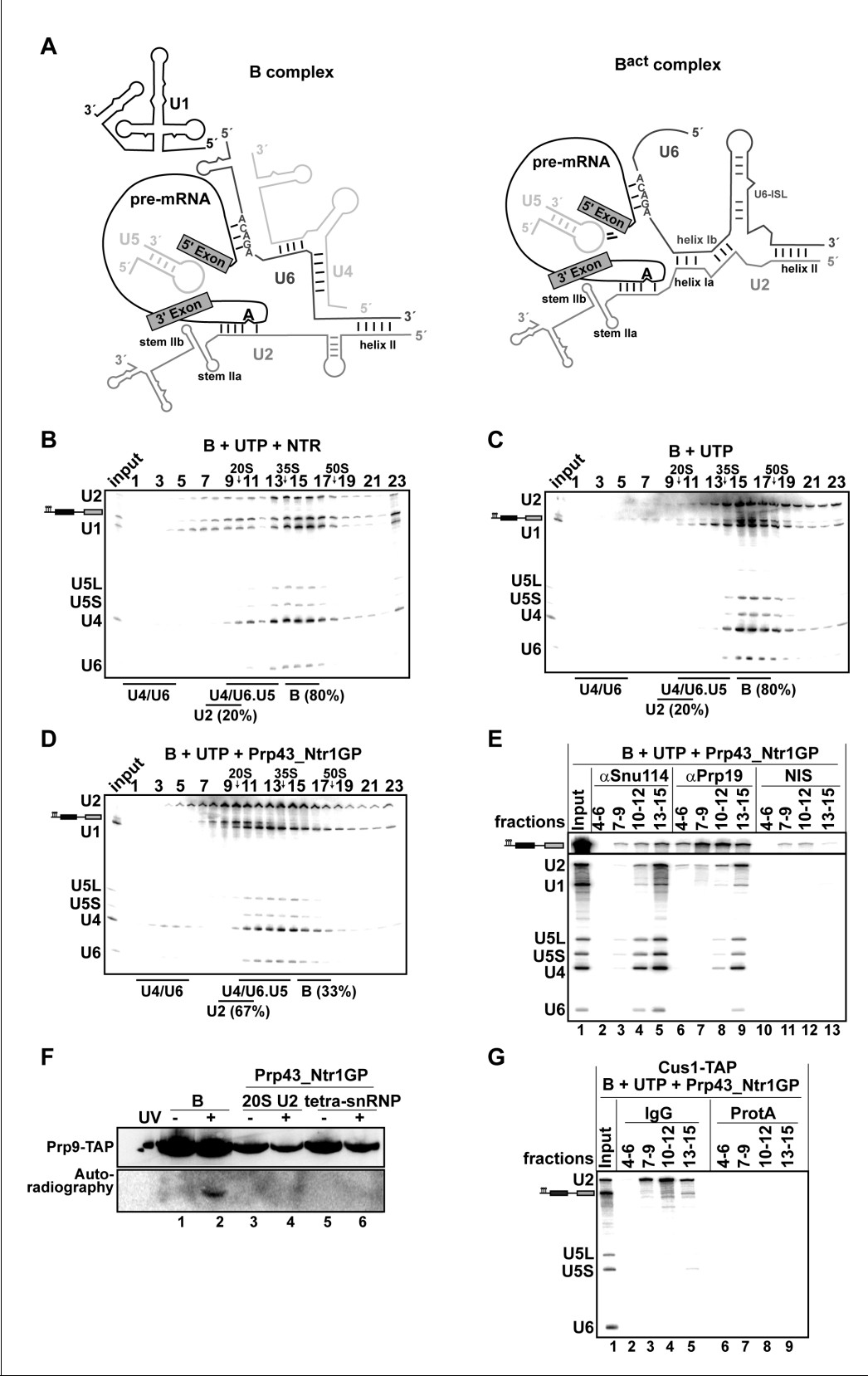

**Figure 3.** Prp43_Ntr1GP disassembles B complexes. (A) Scheme of the RNA network in B and Bact complexes (*Madhani and Guthrie, 1992*). The U6 ACAGA box and U2/U6 helixes I and II are indicated. 10–30% glycerol gradient sedimentation of purified B complexes incubated with UTP and (B) NTR

*Figure 3 continued on next page*

Figure 3 continued

or (C) no recombinant protein or (D) Prp43_Ntr1GP. Samples were analyzed as described in *Figure 1*. Quantification of U2 snRNA released or associated with the B complex (unreleased) was performed as described in *Figure 2*. (E) B complexes were incubated with UTP and Prp43_Ntr1GP and separated on a glycerol gradient. The indicated three gradient fractions were combined and immunoprecipitated with anti-Snu114, anti-Prp19 antibodies or with non-immune serum (NIS). Co-precipitated snRNAs were analyzed on an 8% urea-polyacrylamide gel and identified by Northern blotting using probes that hybridize to U2, U1, U5, U4 and U6 RNA (E) B complexes harboring Prp9-TAP were assembled on uniformly radiolabeled pre-mRNA. One half was left untreated and the other half was disassembled with Prp43_Ntr1GP and UTP, and then analysed on separate glycerol gradients. Peak fractions corresponding to intact B complex (15–18), 20S U2 (9–11) and tetra-snRNP (13–15) were UV-irradiated or untreated. Proteins and protein-RNA crosslinks were analysed as in *Figure 2*. (G) B complexes containing the U2 protein Cus1 tagged with the TAP tag (Cus1-TAP) were incubated with UTP and Prp43_Ntr1GP and separated on a glycerol gradient. The indicated three gradient fractions were combined and immunoprecipitated with IgG Sepharose beads (IgG, lanes 2–5) allowing the selective immunoprecipitation of 20S U2 snRNP (lanes 3,4) or tetra-snRNP (lane 5). The same fractions were also precipitated with Protein A–Sepharose beads (ProtA) (lanes 6–9). Co-precipitated snRNAs were analyzed as above and identified by Northern blotting using probes that hybridize to U2, U5 and U6 RNA.

The following source data is available for figure 3:

**Source data 1.** Protein composition of B complexes disassembled by Prp43_Ntr1GP and UTP.

*Warkocki et al., 2015*). To test whether the 3' end of the intron is also required for displacement of U2 snRNP from the spliceosome by Prp43_Ntr1GP, we used purified $B^{act}$ complexes assembled on an actin pre-mRNA containing only six nucleotides downstream of the BS and lacking the 3' exon (ActinΔ6). This complex cannot be catalytically activated by Prp2 and Spp2 (*Fabrizio et al., 2009*). However, the vast majority of ActΔ6$B^{act}$ complexes were efficiently disassembled in the presence of Prp43_Ntr1GP and ATP, but not ATP alone (*Figure 5A,B*), demonstrating that intron nucleotides downstream of the BS are not required for docking Prp43.

To identify the RNA molecules bound by Prp43_Ntr1GP, we carried out UV-mediated RNA-protein crosslinking experiments (*Figure 6*). For preparative reasons, we used isolated $B^{act\ \Delta Prp2}$ complexes instead of ILSs. Purified $B^{act\ \Delta Prp2}$ complexes assembled on $^{32}$P-labeled Act7-wt pre-mRNA were incubated with 2-fold molar excess of either Prp43_Ntr1GP or as a control Prp43 alone in the absence of ATP (*Figure 6*). Then the $B^{act\ \Delta Prp2}$-Prp43 complexes were separated from excess recombinant protein by glycerol gradient centrifugation and peak fractions containing $B^{act\ \Delta Prp2}$ spliceosomes bound by Prp43 or Prp43_Ntr1GP (as determined by Western blotting; *Figure 6A*, lanes 2,3) were irradiated with UV light at 254-nm, subjected to denaturing conditions to disrupt protein–protein and non-covalent RNA-protein interactions. Finally RNA species crosslinked to Prp43 were immunoprecipitated with anti-Prp43 antibodies and identified by Northern blotting. Prp43_Ntr1GP crosslinked only to pre-mRNA, and not to U2 or any other snRNA (*Figure 6B*, lane 5). Remarkably, Prp43 alone also crosslinked exclusively to the pre-mRNA, though less efficiently than Prp43_Ntr1GP (*Figure 6B*, lanes 3,5).

Taken together our data suggest that Prp43_Ntr1GP interacts with the pre-mRNA in native $B^{act}$ complexes, prior to actively displacing U2 from the BS. Prp43_Ntr1GP is capable of dissociating U2 snRNP in all spliceosomal complexes where it is bound to pre-mRNA including the earliest complex, complex A, consistent with Prp43_Ntr1GP disrupting the U2-BS interaction.

## Prp43_Ntr1GP crosslinks to U2 proteins in the $B^{act}$ complex

Inspection of the 20–30 nt long region upstream of the U2 anchoring site in a variety of yeast pre-mRNA introns did not reveal a conserved sequence motif or higher order structural element that could aid in the specific recruitment of Prp43. Thus, specific protein-protein interactions likely aid in recruiting Prp43 to its docking site on the intron close to the U2 snRNP. To identify Prp43 interacting partners in the spliceosome, we performed protein–protein crosslinking experiments with the homo-bifunctional chemical crosslinker BS3 and $B^{act\ \Delta Prp2}$ complexes incubated with Prp43_Ntr1GP in the absence of ATP and then separated from excess Prp43_Ntr1GP by gradient centrifugation. Crosslinks between Prp43_Ntr1GP and other spliceosomal proteins were identified by MS. Prp43_Ntr1GP was reproducibly crosslinked at multiple lysines to the U2 proteins Cus1 and Hsh155, with crosslinks mapped to residues of the C-terminal OB-fold domain of Prp43 and the G-patch of Prp43_Ntr1GP (*Figure 6—source data 1* and *Figure 6—figure supplement 1*. Additional crosslinks were observed between Prp43_Ntr1GP and the NTC protein Clf1, the RES proteins Bud13 and Pml1, and several

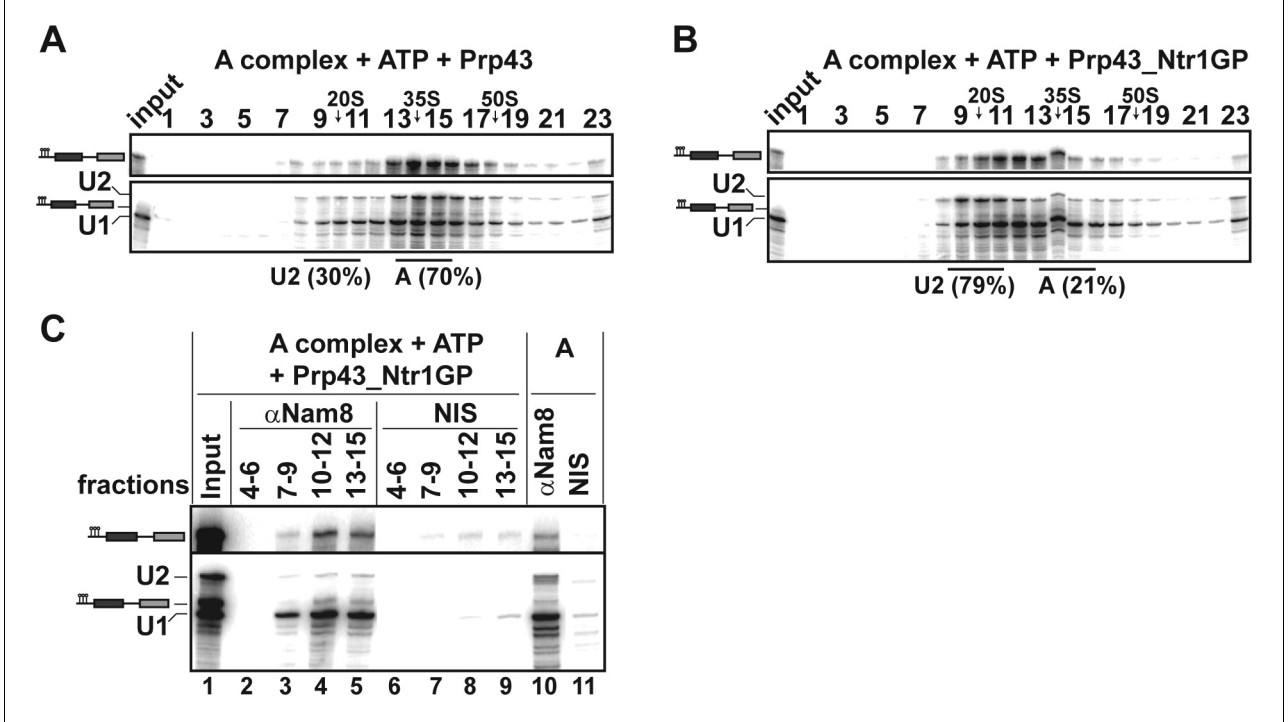

**Figure 4.** Prp43_Ntr1GP disassembles A complexes. 10–30% glycerol gradient sedimentation of purified A complexes incubated with ATP and (**A**) Prp43 or (**B**) Prp43_Ntr1GP. Samples were analyzed as described in *Figure 1* and quantifications were performed as described in *Figure 2*. (**C**) A complexes were incubated with ATP and Prp43_Ntr1GP. Five% of the input was withdrawn (lane 1). Samples were separated on 10–30% glycerol gradients. Every three fractions from 4 to 15 (4–6, 7–9, etc.; panel b) were combined and immunoprecipitated with anti-Nam8 antibodies or with non-immune serum (NIS). In lanes 10–11, immunoprecipitations were performed with purified A complexes.

The following source data is available for figure 4:

**Source data 1.** Protein composition of A complexes disassembled by Prp43_Ntr1GP and ATP.

other splicing factors, albeit in a non-reproducible manner (i.e. in only 1 out of 3 experiments). Taken together, these results suggest that the U2 SF3b proteins Cus1 and Hsh155 help to recruit Prp43_Ntr1GP to the spliceosome and aid in its positioning on the intron near the U2 binding site.

## Discussion

Disassembly of the post-catalytic intron-lariat spliceosome (ILS) is catalyzed by the DEAH-box ATPase Prp43 in cooperation with Ntr2 and the G-patch co-factor Ntr1. However, the target structure(s) of Prp43 in the spliceosome, as well as the mechanism whereby Prp43 is specifically recruited to its target site(s) is not known. To address these questions we initially determined the minimal domain requirements of Ntr1 and Ntr2 to promote disassembly of the ILS by Prp43 in a purified system. A fusion protein of Prp43 and the Ntr1 G-patch (Prp43_Ntr1GP) was as active as the NTR in ILS disassembly and yielded identical snRNP and RNA dissociation products, demonstrating that Prp43 plus the Ntr1 GP are necessary and sufficient, and perform the same function as the NTR during ILS disassembly.

Previous studies indicated that Ntr2 mediates the binding of the NTR to the spliceosome (*Tsai et al., 2005*). However, in our purified system, Ntr2 and the C-terminal domain (CTD) of Ntr1 are not required for Prp43 to bind to its correct target site(s) on the ILS. Moreover, both Prp43 alone and Prp43_Ntr1GP could be crosslinked to the pre-mRNA in B[act] complexes (*Figure 6*), indicating that their recruitment is not dependent on Ntr2 or the CTD of Ntr1. Indeed, although Prp43, Ntr1 and Ntr2 form a trimeric complex, it is not very stable and thus, NTR components can bind independently of one another, as shown previously for Ntr1/Ntr2 (*Chen et al., 2013*) and shown here for

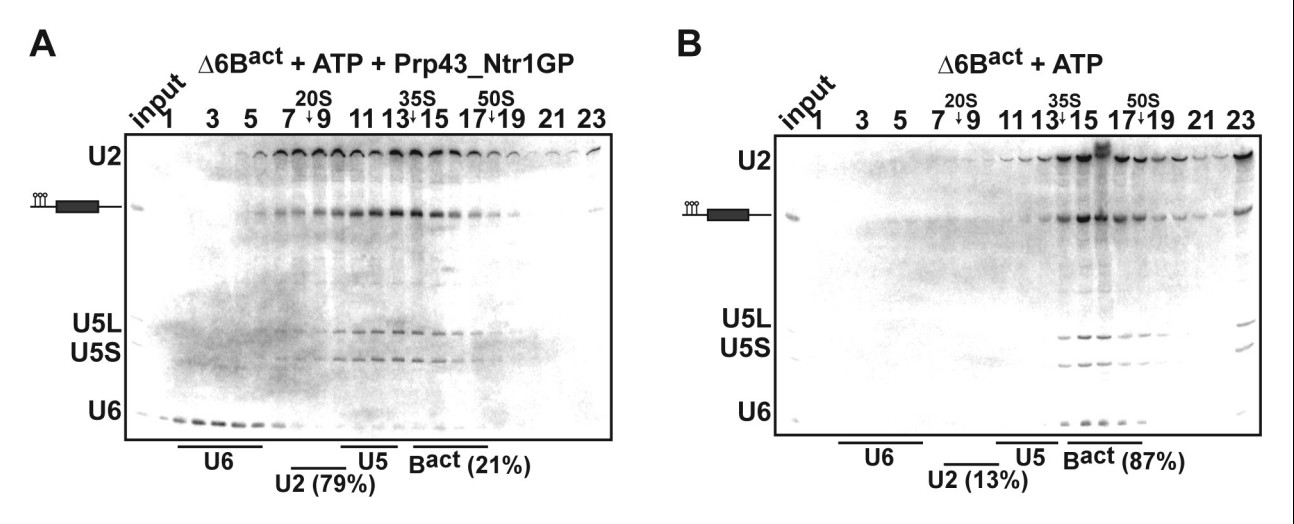

**Figure 5.** Prp43_Ntr1GP disassembles Δ6B$^{act}$ complexes. 10–30% glycerol gradient sedimentation of purified ActΔ6B$^{act}$ spliceosomes assembled on ActinΔ6 pre-mRNAs (*Fabrizio et al., 2009*) and incubated with (A) ATP and Prp43_Ntr1GP or (B) ATP alone. Samples were further processed and quantified as described in *Figures 1* and *2*.

Prp43 and Prp43_Ntr1GP (*Figure 6*). Taken together, our results are most compatible with a model in which Prp43 recognizes its target site(s) in the spliceosome alone (*Figure 6*) and that Ntr1/Ntr2 recognize their spliceosome binding site(s) independently of Prp43 (*Chen et al., 2013*). In the ILS they must bind at neighboring positions to ensure that Ntr1's GP can interact productively with Prp43, leading to ILS disassembly.

Our observation that Prp43_Ntr1GP can dismantle purified wild-type B$^{act}$ complexes efficiently, whereas in the presence of Ntr1/Ntr2 the B$^{act}$ complex is not disassembled by Prp43 (*Figure 2*), indicates that Ntr2 and Ntr1's CTD play an important role in ensuring a productive interaction of Prp43 (i.e. leading to complex disassembly) only with spliceosomes that should be discarded or disassembled. As Prp43 alone can bind the B$^{act}$ complex (*Figure 6*), the lack of disassembly activity by the NTR suggests that Ntr1/Ntr2 do not bind productively or do not have an available binding site yet in the B$^{act}$ complex, and thus Prp43 activity is not stimulated. Accordingly, the Ntr1-bound Ntr2 protein appears to act as a 'doorkeeper', allowing a productive interaction of the Ntr1 GP with Prp43 only when a cognate binding site for Ntr1/Ntr2 exists or is available next to the Prp43-binding site in the spliceosome - as is apparently the case for the ILS.

Our data indicate that the Prp43_Ntr1GP fusion protein is a disassembly factor that recognizes its natural target site on the ILS, but is not under 'access control' by the Ntr1 CTD and Ntr2. Strikingly, not only the pre-catalytic B$^{act}$, but also B and A complexes were dissociated by Prp43_Ntr1GP (*Figures 2*, *3* and *4*). The only structure common to each of these complexes and also the ILS, is the U2 snRNP bound to the intron at/near the BS, suggesting it is the direct target of Prp43. This conclusion was also supported by the observation that (i) intron nucleotides downstream of the BS are not required for docking Prp43_Ntr1GP (*Figure 5*), and (ii) Prp43_Ntr1GP could be crosslinked to U2 SF3b proteins in purified B$^{act}$ complexes (*Figure 6—source data 1*).

Similar to ILS disassembly, in the B$^{act}$, B and A complexes binding of U2 snRNP to the pre-mRNA BS was completely disrupted. The action of Prp43_Ntr1GP not only disrupted the base-pairing interaction between U2 snRNA and the BS, but also interactions between U2 proteins and the intron. In contrast, UV RNA–protein crosslinking, IP experiments and MS showed that in all cases, the U2 SF3a und SF3b proteins remained associated with the U2 snRNA. This indicates that Prp43_Ntr1GP dislodges U2 snRNP from the pre-mRNA BS region as an intact particle and thus that the release of U2 from the BS does not result from prior disassembly of the U2 snRNP.

Prp43 alone preferentially, though inefficiently, unwinds model RNA duplexes in vitro in a 5' to 3' direction (*Tanaka and Schwer, 2006*). The presence of Ntr1 (*Tanaka et al., 2007*) enhances Prp43 activity and allows it to unwind model substrates in both 5' to 3' and 3' to 5' directions. Given that

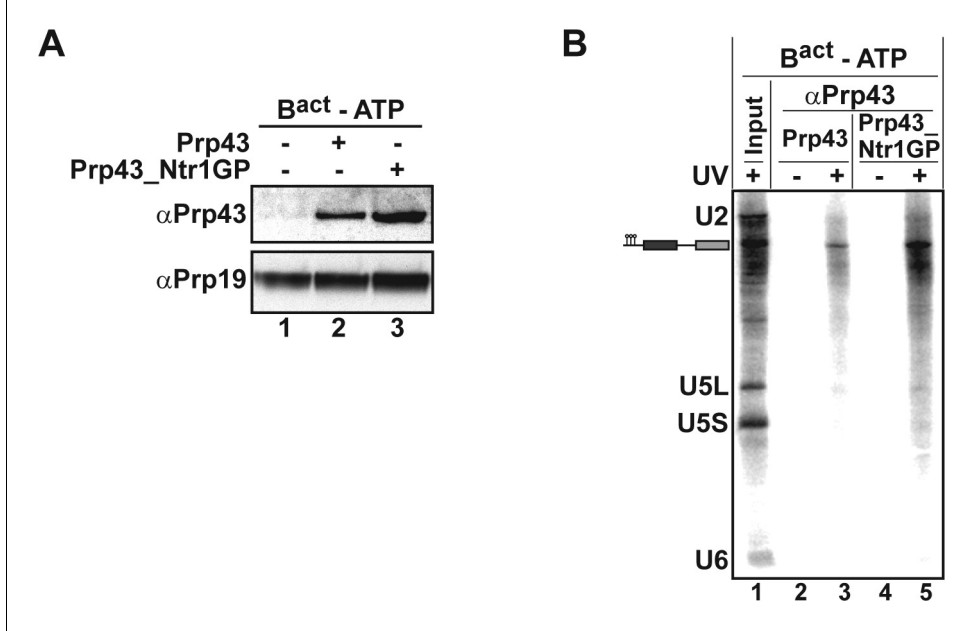

**Figure 6.** Prp43 and Prp43_Ntr1GP bind to the B[act ΔPrp2] complex and interact with the pre-mRNA. (**A**) Purified B[act ΔPrp2] complexes formed on radiolabelled Act7-wt pre-mRNA were incubated without recombinant proteins (lane 1) with Prp43 (lane 2) or Prp43_Ntr1GP (lane 3) in the absence of ATP. Samples were loaded on distinct glycerol gradients. Proteins from the peak fraction were recovered and separated by SDS-PAGE on a 4–12% Bis-TrisNuPAGE polyacrylamide gel (Invitrogen) and visualized by Western blotting using anti-Prp43 or anti-Prp19 antibodies (**B**) Purified B[act ΔPrp2] complexes formed on radiolabelled Act7-wt pre-mRNA were incubated with Prp43 (lanes 2 and 3) or Prp43_Ntr1GP (lanes 4 and 5) in the absence of ATP and the reaction mixture was irradiated with UV light at 254-nm (+) or left untreated (-). Following denaturation, the reaction mixtures were immunoprecipitated with anti-Prp43 antibodies. After proteolytic digestion, coprecipitated snRNAs were identified by Northern blotting using radiolabelled probes that hybridise to U2, U5 or U6 snRNA.

The following source data and figure supplement are available for figure 6:

**Source data 1.** Summary of inter-molecular and intra-molecular protein–protein crosslinking for Prp43_Ntr1GP bound to the B[act ΔPrp2] spliceosome in the absence of ATP.

**Figure supplement 1.** Schematic summary of BS3 crosslinking of Prp43_Ntr1GP with proteins of the B[act ΔPrp2] spliceosome.

---

Prp43 binds the pre-mRNA and the U2 snRNP proteins Cus1 and Hsh155 and can efficiently disassemble ActΔ6Bact complexes (*Figure 5*), our data are consistent with the idea that Prp43_Ntr1GP translocates along the pre-mRNA in a 5' to 3' direction, thereby displacing pre-mRNA-bound U2 proteins and disrupting the U2 snRNA/BS base-pairing interaction. Thus, Prp43 likely acts as a canonical RNPase, most probably coupled with RNA duplex unwinding activity, as shown for the DExH-box RNA helicase NPH-II (*Jankowsky et al., 2000*; *Jankowsky et al., 2001*).

Alternatively, Prp43 could function similarly to DEAD-box proteins, which act locally, disrupting or in some cases promoting RNA structures without significant translocation, as suggested previously (*Guenther and Jankowsky, 2009*; *Jarmoskaite and Russell, 2014*). Indeed DEAH-box proteins that translocate load onto ssRNA and translocate 3' to 5', disrupting RNA structures and displacing proteins as they translocate, as for example Prp2 and Prp22, for which targets have been identified. Prp2 requires nucleotides of pre-mRNA downstream of the BS, suggesting 3' to 5' translocation (*Liu and Cheng, 2012*). Along the way, Prp2 is suggested to remodel the binding of the U2 SF3a/b proteins and a number of others (*Warkocki et al., 2009*; *Lardelli et al., 2010*; *Ohrt et al., 2012*). Likely, destabilization of SF3a/b leads to the exposure of the BS and also to remodeling of the catalytic center of the spliceosome so that RNA elements can interact to initiate the first catalytic reaction. Also the action of Prp22 most likely involves directional movement 3' to 5' on the mature

mRNA (*Wagner et al., 1998*; *Tanaka and Schwer, 2005*; *Schwer, 2008*). Following splicing, Prp22 releases the mRNA from the spliceosome by disrupting base pairs between the U5 snRNP and the mRNA thus breaking several RNA–RNA and RNA–protein contacts (*Company et al., 1991*; *Wagner et al., 1998*; *Tanaka and Schwer, 2005*; *Schwer, 2008*).

In the ILS and B$^{act}$ complex, Prp43_Ntr1GP also disrupted (i) the base-pairing interactions between the U6 and U2 snRNAs (helices I and II), (ii) the base-pairing between the 5' end of the intron and the U6 snRNA ACAGA box, in both cases releasing free U6 snRNA, and (iii) the interaction(s) between the U2 and U5 snRNPs. Thus, Prp43_Ntr1GP could potentially act sequentially on several targets. Alternatively, the U2/BS interaction might serve as a structural 'keystone' that once disrupted by Prp43, destabilizes other interactions in the spliceosome, leading to their passive dissociation. Although we cannot exclude the first possibility, we favor the second, as it is more consistent with the identity of the dissociation products from the various complexes. That is, U2 snRNP is completely displaced from the BS when the B complex is disassembled by Prp43_Ntr1GP, but one third of the U2 snRNP remains stably bound to the tri-snRNP in the form of a U2.U4/U6.U5 tetra-snRNP. Binding of U2 snRNP to the tri-snRNP is stabilized in part by the U2/U6 helix II base pairing interaction. If this helix, which also exists in the B$^{act}$ complex and in the ILS, were a target for Prp43_Ntr1GP, then one would expect complete dissociation of the tetra-snRNP into tri-snRNP and U2 snRNP – which however does not occur. Moreover, the U6 snRNA remains base-paired with the U4 snRNA, both in the tri-snRNP and in the tetra-snRNP (*Figure 3B*). Finally, after dissociation of B$^{act}$, the U5 snRNP remains stably bound to the pre-mRNA (*Figures 2C,D*). Likewise, the binding of U1 snRNP to the 5'SS is not disrupted by Prp43 after dislodging U2 snRNP from complex A (*Figure 4C*). The latter observations confirm that Prp43_Ntr1GP is not a promiscuous disassembly enzyme. Taken together, our results support the idea that the U2 snRNP/BS interaction is the principal target of Prp43 in the spliceosome and that all other Prp43-mediated spliceosome dissociation products are generated as a consequence of the dissociation of this main building block.

## Materials and methods

### Spliceosome purification and reconstitution

Yeast A, B, B$^{act\ \Delta Prp2}$ or ActΔ6B$^{act}$ complexes were assembled in yeast whole cell extracts by incubating with the indicated pre-mRNA containing MS2 aptamers at 23°C for 45 min. Samples were centrifuged for 10 min at 9000 rpm and loaded onto columns containing 200–450 µl of amylose matrix equilibrated with GK75 buffer (20 mM HEPES-KOH pH 7.9, 1.5 mM MgCl$_2$, 75 mM KCl, 5% glycerol, 0.01% NP40, 0.5 mM DTT). The matrix was washed twice with 10 ml GK75 buffer. Spliceosomes were eluted with 12 mM maltose in GK75 buffer and 400 µl was loaded onto linear 10–30% (v/v) glycerol gradients containing GK75 buffer. Samples were centrifuged for 16 hr at 21,500 rpm in a TH660 rotor (Thermo Scientific, Waltham, MA) and harvested manually from the top in 23 fractions of 175 µl. Fractions were analyzed by Cherenkov counting in a scintillation counter. Peak fractions containing complexes were pooled and the glycerol concentration was adjusted to 5–10% with GK75 buffer without glycerol. ILSs were obtained as described previously (*Fourmann et al., 2013*). To assemble A complexes, extract was prepared from the modified yeast strain W303a *MAT*a [*leu2-3,112 trp1-1 can1-100 ura3-1 ade2-1 his3-11,15*] [*phi⁺*], *HisMX6-$_{PGAL1}$-3HA-PRP19*. Prp19 was metabolically depleted by growth in medium containing glucose (1% yeast extract, 2% peptone, 2% glucose) for 16 hr at 30°C. Yeast B and B$^{act\ \Delta Prp2}$ spliceosomal complexes were assembled in heat-inactivated extracts from the yeast strain *prp2-1* (*Yean and Lin, 1991*) or wild-type extract from the yeast strain BJ2168 for ActΔ6B$^{act}$ (*Fabrizio et al., 2009*), and were purified essentially as described in *Fabrizio et al. (2009)*, *Warkocki et al. (2009)*, *Fourmann et al. (2013)*. Before splicing, Act7-wt, or ActinΔ6 pre-mRNAs were incubated with a thirty-fold molar excess of purified MS2-MBP fusion protein at 4°C for 30 min in 20 mM HEPES-KOH (pH 7.9). Typically, a 6–200 ml splicing reaction containing 1.8 nM of $^{32}$P-labeled Act7-wt pre-mRNA (specific activity 2–500 cpm/fmol) was performed in 62.5 mM KPO$_4$ (pH 7.4), 3% PEG 8000, 2.5 mM MgCl$_2$, 2.0 mM ATP for A, B$^{act\ \Delta Prp2}$ and ActΔ6B$^{act}$ complexes, and 0.05 mM ATP for B complexes, 2.0 mM spermidine, and 40% yeast extract in buffer D [20 mM HEPES-KOH, pH 7.9, 50 mM KCl, 0.2 mM EDTA pH 8.0, 20% (v/v) glycerol, 0.5 mM DTT, and 0.5 mM PMSF].

To obtain intron-lariat spliceosomes, $B^{act\ \Delta Prp2}$ complexes bound to the amylose matrix were supplemented with a ten-fold molar excess of recombinant proteins (Prp2, Spp2, Cwc25, Prp16, Slu7 and Prp18) and the reaction volume was adjusted to 400 µl with GK75 buffer; then 40 µl of 10× 'rescue' solution [200 mM $KPO_4$ (pH 7.4), 10 mM $MgCl_2$, 20 mM ATP, 10% PEG 8000] were added to the reaction. After thorough mixing, the reaction was incubated at 23°C for 45 min. Matrixes were subsequently washed 3 times with 10 column volumes of GK75 buffer. Then a ten-fold molar excess of recombinant Prp22 was added and the volume was adjusted to 400 µl with 1× 'rescue' solution prepared in GK75 buffer. After thorough mixing, the reaction was incubated at 23°C for 10 min. The supernatant (containing the released ILS) was collected, and GK75 buffer was added to the matrix to a final volume of 400 µl. After gentle mixing and repeated centrifugation for 1 min at 2000 rpm the supernatant was collected and pooled with the first supernanant.

## Spliceosome disassembly assays

To dismantle spliceosomes (A, B, $B^{act\ \Delta Prp2}$, ILS or ActΔ6$B^{act}$), samples were incubated with distinct combinations of a ten-fold molar excess of recombinant Prp43, the dimer (Ntr1, Ntr2,), Prp43_Ntr1GP or Ntr2, and the volume was adjusted to 400 µl with 1x 'rescue' solution prepared in GK75 buffer containing ATP or UTP (B complexes only). After thorough mixing, the mixture was incubated at 23°C for 10 min and then subjected to glycerol gradient centrifugation for 2 hr at 60,000 rpm in a TH660 rotor (Thermo Scientific) and harvested manually from the top in 23 fractions of 175 µl. For mass spectrometry or immunoprecipitation experiments, every two or three fractions were pooled. For RNA analysis, each fraction was subjected to digestion with Proteinase K followed by phenol-chloroform-isoamyl alcohol (PCI) extraction. RNA was precipitated with ethanol, and then analyzed by PAGE on 8% polyacrylamide, 8M urea gels (PAGE) and visualized by autoradiography or Northern blot analysis.

## Cloning and expression strategy

Recombinant Prp2, Spp2, Cwc25, Prp16, Slu7, Prp18, Prp22, Prp43, and dimer (Ntr1, Ntr2) were purified as described in *Fourmann et al., (2013)*. Prp43_Ntr1GP was prepared as described in *Christian et al. (2014)*.

## BS3 protein–protein crosslinking

$B^{act\ \Delta Prp2}$ complexes were eluted from amylose beads as described above. Remaining ATP was depleted by addition of 2U hexokinase and 12 mM glucose and by incubating for 10 min at 23°C. 330 nM Prp43_Ntr1GP was added to the sample in the absence of ATP and the mixture was incubated for 5 min at 23°C, followed by 5 min at 4°C, and subsequently loaded on a 10–30% (v/v) glycerol gradient containing GK75 buffer. Samples were centrifuged for 16 hr at 25,000 rpm in a Surespin 630 rotor (Thermo Scientific) and harvested manually from the top in 29 fractions of 555 µl. Fractions were analyzed by Cherenkov counting. Peak fractions containing $B^{act\ \Delta Prp2}$ complexes with Prp43_Ntr1GP bound were pooled and contained ∼50–100 pmol of spliceosomes. The sample was incubated with 150 µM BS3, a homobifunctional amino-specific chemical crosslinker with a spacer length of 11 Å, for 30 min at 24°C. After quenching with Tris-HCl, complexes were pelleted by ultracentrifugation in S100-AT4 rotor (Thermo Scientific) for 3 hr and analyzed essentially as described in *Leitner et al. (2014)* with the following modifications: precipitated material was dissolved in 4 M urea/50 mM ammonium bicarbonate, reduced with DTT, alkylated with iodoacetamide, diluted to a final concentration of 1M urea and digested with trypsin (1:20 w:w). Peptides were reverse-phase extracted and fractionated by gel filtration on a Superdex Peptide PC3.2/30 column (GE Healthcare, Little Chalfont, UK). The elution volume of 1.2–1.8 ml was collected in 50 µl fractions, which were analyzed on Thermo Orbitrap Fusion Tribrid (*Figure 6—source data 1a and b* experiment 1) or Thermo Q Exactive (*Figure 6—source data 1a and b* experiments 2 and 3) mass spectrometers. Protein–protein crosslinks were identified by pLink1.22 search engine and filtered at FDR 1% according to the recommendations of the developers (*Yang et al., 2012*).

## Mass spectrometry

For mass-spectrometric identification of proteins associated with the various gradient-fractionated spliceosomal complexes, every two fractions of the gradient containing $B^{act\ \Delta Prp2}$ dismantled by

Prp43_Ntr1GP and every three fractions of the gradient containing A or B complexes dismantled by Prp43_Ntr1GP were pooled in low-protein-binding reaction tubes (Eppendorf, Hamburg, DE). 300 µl of the pooled fractions were supplemented with 40 µg glycoblue, 30 µl of 3 M NaOAc (pH 5.2) and 1 ml of ethanol. After thorough mixing, the solution was precipitated overnight at –80°C and then centrifuged for 30 min at 13,000 rpm and 4°C in a Microfuge. Subsequently, the pellet was washed with 70% ethanol, dried in a vacuum dryer and re-suspended in 1× SDS-PAGE loading buffer (Invitrogen) and heated to 70°C for 10 min. Proteins recovered from the Prp43-dismantled complexes were separated by SDS-PAGE on a 4–12% Bis-Tris NuPAGE polyacrylamide gel (Invitrogen, Carlsbad, CA) and stained with Coomassie blue. Entire lanes were cut into 23 slices, and proteins were digested in-gel with trypsin and extracted as described previously (*Shevchenko et al., 1996*). Resulting peptides were analyzed in an LTQ Orbitrap XL (ThermoFisher Scientific) mass spectrometer under standard conditions. Proteins were identified by searching fragment spectra against the S. Cerevisiae Genomic Database (SGD) using Mascot as a search engine.

## UV crosslinking of spliceosomes

Approximately 0.5 to 2 pmol of purified yeast $B^{act\ \Delta Prp2}$ complexes were incubated with a 2-fold molar excess of either Prp43_Ntr1GP or Prp43 alone in the absence of ATP, pipetted onto pre-cooled parafilm and then irradiated for 45 s with UV light at 254 nm on ice (*Urlaub et al., 2002*). To the irradiated and non-irradiated control samples of spliceosomal complexes, SDS was added to a final concentration of 1%, incubated for 15 min at 70°C and allowed to cool to room temperature before the addition of Triton X-100 to a final concentration of 2%. The samples were then diluted with 10 volumes of NET-150 buffer (50 mM Tris-HCl pH 7.4, 150 mM NaCl, 0.05% NP-40) and subjected to immunoprecipitation for 2 hr at 4°C.

## Immunoprecipitation of disassembled spliceosomes and of RNA–protein crosslinks

Protein A Sepharose or Protein A Sepharose beads that carried IgG (IgG-Sepharose) (GE Healthcare) resins were washed with NET-150 buffer and incubated at 4°C with UV-crosslinked or non-crosslinked spliceosomal complexes prepared as above. After washing with NET-150 buffer 3 times, beads were incubated with 2-times PK-mix (1 mg/ml Proteinase K, 200 mM Tris-HCl pH 7.4, 300 mM NaCl, 25 mM EDTA pH 8.0, 1% SDS) for 30 min at 37°C by mixing at 800 rpm, followed by PCI extraction, chloroform extraction and ethanol precipitation. The pellet was dried and dissolved in water.

## Acknowledgements

We thank T Conrad, M Raabe and U Plessmann for excellent technical assistance and KL Boon for Prp19-depleted extracts. We thank C L Will for helpful comments on the manuscript.

## Additional information

### Funding

| Funder | Grant reference number | Author |
| --- | --- | --- |
| Deutsche Forschungsgemeinschaft | SFB 860 | Ralf Ficner Reinhard Lührmann |

The funders had no role in study design, data collection and interpretation, or the decision to submit the work for publication.

### Author contributions

J-BF, Conception and design, Acquisition of data, Analysis and interpretation of data, Drafting or revising the article; OD, DEA, MJT, HU, Conception and design, Acquisition of data, Analysis and interpretation of data; RF, Conception and design, Analysis and interpretation of data, Contributed unpublished essential data or reagents; PF, Conception and design, Analysis and interpretation of

data, Drafting or revising the article, Contributed unpublished essential data or reagents; RL, Conception and design, Analysis and interpretation of data, Drafting or revising the article

## Author ORCIDs

Reinhard Lührmann, http://orcid.org/0000-0002-6403-4432

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
