## [Decision Letter]

Thank you for submitting your article "The target of the DEAH-box NTPase Prp43 in *S. cerevisiae* spliceosomes is the U2 snRNP-intron interaction" for consideration by *eLife*. Your article has been favorably evaluated by James Manley (Senior editor) and three reviewers, one of whom, (Timothy Nilsen) is a member of our Board of Reviewing Editors.

The reviewers have discussed the reviews with one another and the Reviewing Editor has drafted this decision to help you prepare a revised submission.

Summary:

This paper by Lührmann and colleagues presents a series of experiments aimed at elucidating the target of Prp43 (a DEAH box RNA helicase) in the disassembly of the spliceosome following catalysis. Prp43 mediated spliceosome disassembly is catalyzed in the context of a complex between Prp43 and two other proteins, Ntr1 and Ntr2. A surprising finding here is that a Prp43-Ntr1 G-patch fusion protein can function in disassembly in the absence of Ntr1 and 2. Using this fusion protein the authors show through a careful set of experiments that the target of Prp43's helicase activity is the U2 snRNP/pre-mRNA branch site interaction. In essence Prp43 acts as an RNPase to dissociate U2 snRNP from the pre-mRNA. The fusion protein is capable of dissociating U2 snRNP in all complexes where it is bound to pre-mRNA including the earliest complex, complex A.

Essential revisions:

1) After discussion the reviewers agreed that the data presented in Figure 6 were not compelling. These data could be made more convincing by repeating the experiments using pre-mRNAs containing site-specific labels. Alternatively, the Figure could be deleted along with the discussion of it in the text and the conclusions softened appropriately.

2) It was felt that the results presented in this manuscript should be discussed in light of studies by others that have identified the targets of Prp2 and Prp22.

*Reviewer #1:*

The significance of this work goes well beyond the pre-mRNA splicing field in that it provides the first example (at least in my knowledge) of a specific target for an RNA helicase, a large family of enzymes that function in all aspects of cellular RNA metabolism.

The data are of exceptional quality and they are presented clearly and appropriately discussed. I recommend acceptance without significant revision in *eLife*.

*Reviewer #2:*

Previous work from the Lührmann and Ficner labs established that the NTR complex (composed of Prp43, Ntr1 and Ntr2) is necessary and sufficient for spliceosome recycling by promoting the dissociation of intron-lariat spliceosomes (ILS), and that a fusion protein between the DEAH-box NTPase Prp43 and part of the Glycine patch domain of Ntr1 displays elevated RNA-stimulated ATPase and RNA helicase activities, suggesting that this domain of Ntr1 activates Prp43 function. Fourmann et al. now report a detailed study of the biochemical properties of the Prp43_Ntr1GP fusion, and show that 1) the fusion protein promotes ILS disassembly as efficiently as the complete NTR complex; 2) the fusion protein promotes efficient disassembly of stalled B^actΔPrp2^, B or A complexes, activities that the NTR complex do not display; 3) the fusion protein does not require pre-mRNA sequences downstream from the branch point (BPS) (which are however required for Prp2 helicase-mediated spliceosome activation), while RNA-protein and protein-protein crosslinking suggest that Prp43_Ntr1GP is targeted to the pre-mRNA through interaction with U2 snRNP components and contacts with pre-mRNA sequences upstream of the BPS.

Collectively, the results nicely support the notion that Prp43 is activated by the GP domain of Ntr1, promoting the disassembly of various spliceosomal complexes, most likely by targeting -at least initially- the interaction between U2 snRNP and the pre-mRNA. These results have general implications for the targeting mechanisms of DEAH box helicases and for how their associated factors can modulate the timing and specificity of their activities.

In my opinion the manuscript could be improved if the following issues would be addressed/further discussed:

1) The experiments are generally well-controlled and the results are crisp and convincing. However, while one supplementary figure of the previous Fourmann et al. G&D 2013 paper documented low intrinsic ILS disassembly activity of Prp43 in the absence of Ntr proteins, for the sake of completeness it would be useful to show a direct comparison between Prp43_Ntr1GP and Prp43 in Figure 1. A more intriguing question, relevant to the mechanism of helicase activation, is the extent to which the effect of the Glycine Patch is specific of Ntr1's domain or whether any other Glycine Patch would work if similarly fused to Prp43.

2) Figure 2: the conclusion that U2 snRNP is preferentially dissociated from B^act^ complex is based upon a) closer sedimentation profile of pre-mRNA with other snRNAs and b) lack of immunoprecipitation of U2 components using antibodies against components of other complexes (Snu114, Prp19). Can the authors rule out that more than one dissociation pathway/intermediate exists and that in a subset of complexes, Prp43_Ntr1GP may dissociate first other snRNPs? To assess this, it could be helpful to carry out immunoprecipitation assays with antibodies against U2 snRNP components, to validate the quantitative disassembly of the particle from the pre-mRNA while other snRNPs stay. While the result of Figure 2 indeed suggests decreased interaction of Hsh155 with the pre-mRNA, the lower levels of the protein in the Prp4343_Ntr1GP-induced complexes, together with the relatively low autoradiography signal, makes it difficult to judge whether the interaction is really absent or just decreased in proportion to the levels of Hsh155 in the complex.

3) A similar caveat could apply to the result of Figure 3: given the relatively weak RNA crosslinking signal observed for complex B (despite the high levels of Prp9-TAP in the complex), it is unclear whether a proportionally equivalent autoradiography signal corresponding to the substantially lower levels of Prp9-TAP present in the other complexes would be detectable in this assay.

4) Figure 4: as for Figure 2, immunoprecipitation with anti-U2 components could be helpful to validate their quantitative disassembly from pre-mRNAs.

5) Figure 6: given the pattern of RT stops induced by UV irradiation of purified in vitro transcribed RNA (lane 14), it is not easy to see how the very similar pattern observed in the presence of complexes and Prp43_Ntr1GP can unambiguously prove that these are sites of direct interaction with this protein.

*Reviewer #3:*

The study demonstrates that the Prp43_Ntr1GP fusion protein, characterized previously by Christian et al., 2014, can disassemble intron-lariat containing splicing complexes (ILS) as efficiently as the NTR (Prp43/Ntr1/Ntr2). Importantly, the products of the reaction are the same – IL RNA, U6 snRNA and U2 and U5 snRNPs. The authors go on to show that Prp43_Ntr1GP can also dissociate U2 snRNP from pre-mRNA in purified A, B and B^act^ spliceosomes, which are normally not substrates for disassembly by NTR [except at specific stages of the splicing pathway (Chen et al., 2013)]. A common element in all the cases is the U2 snRNP engaged at the branchpoint and the authors suggest a model wherein Prp43_Ntr1GP binds in the intron upstream of the branchsite and translocates in the 5' to 3' direction to displace the U2 snRNP. The authors show that Prp43_Ntr1GP and, with greatly reduced efficiency, Prp43 can be UV cross-linked to pre-mRNA in purified B^act^ spliceosomes and using primer extension assays, they map crosslinks in the RNA. The basis for concluding that the crosslinks in the intron (-33 to -25) indicates Prp43_Ntr1GP binding at this position is unclear. The same primer extension stops are present in the Input lanes, independent of whether or not Prp43_Ntr1GP was added and enrichment upon immunoprecipitation with anti-Prp43 only means that Prp43_Ntr1GP is associated with the complex (IP for any of the U2 proteins would likely yield the same picture). The argument that 'no other proteins are known to interact with this region of the intron' (Results) seems rather weak without a reference to data substantiating this statement. Moreover, doesn't the finding that Prp43_Ntr1GP cross-linked more efficiently to the pre-mRNA than Prp43 indicate that binding might be via the G-patch? (Christian et al., 2014 mapped an RNA binding site within the Ntr1 G-patch).

In summary, the finding that Prp43_Ntr1GP suffices to disassemble ILS in a purified system is very interesting and the model is plausible (although there is not a strong basis for proposing translocation – well-characterized DExH proteins that translocate do so in a 3' to 5' fashion). However, the data (using the B^act^ complex, which is not a natural substrate for disassembly by NTR) shown in support of Prp43's interaction with the intron nucleotides upstream of the branchsite are, in my opinion, not convincing.

---

## [Author Response]

*Essential revisions: 1) After discussion the reviewers agreed that the data presented in Figure 6 were not compelling. These data could be made more convincing by repeating the experiments using pre-mRNAs containing site-specific labels. Alternatively, the Figure could be deleted along with the discussion of it in the text and the conclusions softened appropriately.* As suggested by the reviewing editors, in our revised manuscript we have deleted Figure 6 along with the discussion of it in the text. In addition, the conclusions have been softened appropriately as also two of the reviewers had the opinion that the data presented in Figure 6 were not convincing. In our revised manuscript a new main Figure 6 was included with the experiments of the previous Figure 6—figure supplement 1, which shows that Prp43 alone and Prp43_Ntr1GP crosslink only to pre-mRNA in the B^act ΔPrp2^ complex, and not to U2 or any other snRNAs. The experiment is described in the Results section (subsection “Prp43_Ntr1GP interacts only with the pre-mRNA and not with U2 or any other snRNA in the B^act^ complex”, second paragraph).

*2) It was felt that the results presented in this manuscript should be discussed in light of studies by others that have identified the targets of Prp2 and Prp22.* In our revised manuscript we have added a new paragraph in which we describe also the spliceosomal targets of Prp2 and Prp22 and discuss possible differences between Prp2 and Prp22 that translocate in a 3' to 5' direction and Prp43 which may move along the intron in a 5’ to 3’ direction or may even act locally by disrupting RNA structures without significant translocation, as suggested previously by Guenther and Jankowsky (Guenther UP, Jankowsky E. 2009. Helicase multitasking in ribosome assembly. Mol Cell 36: 537-538) and also indicated by reviewer #3 (Discussion, sixth paragraph).

Reviewer #2:

We have revised Figure 1 and added an additional panel (panel D), which shows a direct comparison between Prp43_Ntr1GP and Prp43, along the lines recommended by the reviewer.

*A more intriguing question, relevant to the mechanism of helicase activation, is the extent to which the effect of the Glycine Patch is specific of Ntr1's domain or whether any other Glycine Patch would work if similarly fused to Prp43.* This is indeed a very interesting question, which will be addressed in our future studies.

*2) Figure 2: the conclusion that U2 snRNP is preferentially dissociated from B^act^ complex is based upon a) closer sedimentation profile of pre-mRNA with other snRNAs and b) lack of immunoprecipitation of U2 components using antibodies against components of other complexes (snu114, Prp19). Can the authors rule out that more than one dissociation pathway / intermediate exists and that in a subset of complexes, Prp43_Ntr1GP may dissociate first other snRNPs? To assess this, it could be helpful to carry out immunoprecipitation assays with antibodies against U2 snRNP components, to validate the quantitative disassembly of the particle from the pre-mRNA while other snRNPs stay. While the result of Figure 2 indeed suggests decreased interaction of Hsh155 with the pre-mRNA, the lower levels of the protein in the Prp4343_Ntr1GP-induced complexes, together with the relatively low autoradiography signal, makes it difficult to judge whether the interaction is really absent or just decreased in proportion to the levels of Hsh155 in the complex.* In our revised manuscript, to validate the quantitative dissociation of the U2 snRNP from the pre-mRNA, we carried out IP experiments using the Cus1-TAP extract. Disassembled B^act ΔPrp2^ spliceosomes were incubated with Protein A–Sepharose beads that carried IgG, to which the TAP tag binds efficiently. We show a selective immunoprecipitation of the U2 snRNP, indicating quantitative dissociation of the U2 particle from the pre-mRNA, which is thus not coprecipitated together with the U2 snRNP (Figure 2, subsection “Prp43_Ntr1GP, but not the NTR, disassembles B^act^ complexes”, second paragraph).

*3) A similar caveat could apply to the result of Figure 3: given the relatively weak RNA crosslinking signal observed for complex B (despite the high levels of Prp9-TAP in the complex), it is unclear whether a proportionally equivalent autoradiography signal corresponding to the substantially lower levels of Prp9-TAP present in the other complexes would be detectable in this assay.* Similarly, in our revised manuscript we show new pull-down experiments using IgG-Sepharose and the Cus1-TAP extract, which confirmed that after disassembly of the B complex by Prp43_Ntr1GP, the pre-mRNA is not coprecipitated together with the U2 snRNP (Figure 3, lanes 3,4), and it is only slightly precipitated from fractions 13-15 containing the U2.U4/U6.U5 tetra-snRNPs and some U2 and U4/U6.U5 tri-snRNPs (lane 5)(subsection “The U2 snRNP-intron interaction is a major target of Prp43 in the spliceosome”, last paragraph).

*4) Figure 6: given the pattern of RT stops induced by UV irradiation of purified* in vitro

*transcribed RNA (lane 14), it is not easy to see how the very similar pattern observed in the presence of complexes and Prp43_Ntr1GP can unambiguously prove that these are sites of direct interaction with this protein.* In our revised manuscript we have deleted Figure 6, the primer extension assays, as the reviewing editors and the reviewers had the opinion that the data presented in Figure 6 were not convincing. Please see our answer to comment 1 of the reviewing editors.

Reviewer #3:

The study demonstrates that the Prp43_Ntr1GP fusion protein, characterized previously by Christian et al., 2014, can disassemble intron-lariat containing splicing complexes (ILS) as efficiently as the NTR (Prp43/Ntr1/Ntr2). […] The argument that 'no other proteins are known to interact with this region of the intron' (Results) seems rather weak without a reference to data substantiating this statement. Moreover, doesn't the finding that Prp43_Ntr1GP cross-linked more efficiently to the pre-mRNA than Prp43 indicate that binding might be via the G-patch? (Christian et al., 2014 mapped an RNA binding site within the Ntr1 G-patch).

In our revised manuscript we have deleted Figure 6, the primer extension assays, as the reviewing editors and the reviewer #2 had the opinion that the data presented in Figure 6 were not convincing. Please see our answer to comment 1 of the reviewing editors.

In summary, the finding that Prp43_Ntr1GP suffices to disassemble ILS in a purified system is very interesting and the model is plausible (although there is not a strong basis for proposing translocation – well-characterized DExH proteins that translocate do so in a 3' to 5' fashion). However, the data (using the B^act^ complex, which is not a natural substrate for disassembly by NTR) shown in support of Prp43's interaction with the intron nucleotides upstream of the branchsite are, in my opinion, not convincing.

We agree that there is not a strong basis for proposing translocation and that other characterized DExH proteins that translocate do so in a 3' to 5' fashion. For this reason we added a new paragraph in our Discussion were we point out that “Prp43 could function similarly to DEAD-box proteins, which act locally, disrupting or in some cases promoting RNA structures without significant translocation, as suggested previously (Guenther and Jankowsky 2009; Jarmoskaite and Russell 2014). Indeed DEAH-box proteins that translocate load onto ssRNA and translocate 3’ to 5’, disrupting RNA structures and displacing proteins as they translocate, as for example Prp2 and Prp22, for which targets have been identified.”